VERO cells harbor a poly-ADP-ribose belt partnering their epithelial adhesion belt

Lafon-Hughes Laura 1 lauralafon2010@gmail.com
Vilchez Larrea Salomé C. 2
Kun Alejandra 3 4
Fernández Villamil Silvia H. 2 5 s.villamil@ingebi.conicet.gov.ar
1 Departamento de Genética, Instituto de Investigaciones Biológicas Clemente Estable (IIBCE) , Montevideo , Uruguay
2 Instituto de Investigaciones en Ingeniería Genética y Biología Molecular “Dr. Héctor N. Torres”, Consejo Nacional de Investigaciones Científicas y Técnicas, Ciudad Autónoma de Buenos Aires , Argentina
3 Departamento de Proteínas y Ácidos Nucleicos, Instituto de Investigaciones Biológicas Clemente Estable (IIBCE) , Montevideo , Uruguay
4 Departamento de Biología Celular y Molecular, Sección Bioquímica, Facultad de Ciencias, Universidad de la República , Montevideo , Uruguay
5 Departamento de Química Biológica, Facultad de Farmacia y Bioquímica, Universidad de Buenos Aires, Ciudad Autónoma de Buenos Aires , Argentina
Kim Cheorl-Ho
Electronic publication date: 2014 Oct 14
Publication date: 2014
Volume: 2
Electronic Location ID: e617
Received 2014 Jun 26; Accepted 2014 Sep 22
Copyright: © 2014 Lafon-Hughes et al.
Copyright year: 2014
Copyright holder: Lafon-Hughes et al.
License: This is an open access article distributed under the terms of the Creative Commons Attribution License, which permits unrestricted use, distribution, reproduction and adaptation in any medium and for any purpose provided that it is properly attributed. For attribution, the original author(s), title, publication source (PeerJ) and either DOI or URL of the article must be cited.
License URL: https://creativecommons.org/licenses/by/4.0/

Keywords: Tankyrase, PARP, PAR, Actin, Vinculin, E-cadherin, Alpha-catenin, Adherens junctions, XAV 939, PJ34

Funding: Consejo Nacional de Investigaciones Científicas y Técnicas Universidad de Buenos Aires Agencia Nacional de Promoción Científica y Tecnológica Fundación Florencio Fiorini Programa de Desarrollo de las Ciencias Básicas This work was supported by Consejo Nacional de Investigaciones Científicas y Técnicas (CONICET, Argentina); Universidad de Buenos Aires (Argentina), Agencia Nacional de Promoción Científica y Tecnológica (Argentina), Fundación Florencio Fiorini (Argentina), and Programa de Desarrollo de las Ciencias Básicas (PEDECIBA, Uruguay). The funders had no role in study design, data collection and analysis, decision to publish, or preparation of the manuscript.

==============================
Poly-ADP-ribose (PAR) is a polymer of up to 400 ADP-ribose units synthesized by poly-ADP-ribose-polymerases (PARPs) and degraded by poly-ADP-ribose-glycohydrolase (PARG). Nuclear PAR modulates chromatin compaction, affecting nuclear functions (gene expression, DNA repair). Diverse defined PARP cytoplasmic allocation patterns contrast with the yet still imprecise PAR distribution and still unclear functions. Based on previous evidence from other models, we hypothesized that PAR could be present in epithelial cells where cadherin-based adherens junctions are linked with the actin cytoskeleton (constituting the adhesion belt). In the present work, we have examined through immunofluorescence and confocal microscopy, the subcellular localization of PAR in an epithelial monkey kidney cell line (VERO). PAR was distinguished colocalizing with actin and vinculin in the epithelial belt, a location that has not been previously reported. Actin filaments disruption with cytochalasin D was paralleled by PAR belt disruption. Conversely, PARP inhibitors 3-aminobenzamide, PJ34 or XAV 939, affected PAR belt synthesis, actin distribution, cell shape and adhesion. Extracellular calcium chelation displayed similar effects. Our results demonstrate the existence of PAR in a novel subcellular localization. An initial interpretation of all the available evidence points towards TNKS-1 as the most probable PAR belt architect, although TNKS-2 involvement cannot be discarded. Forthcoming research will test this hypothesis as well as explore the existence of the PAR belt in other epithelial cells and deepen into its functional implications.

Introduction

Poly-ADP-ribose (PAR) is a linear or branched polymer of up to 400 ADP-ribose units that binds (covalently or not) to target proteins. PAR is synthesized by poly-ADP-ribose-polymerases (PARPs) and the key catabolic enzyme is poly-ADP-ribose-glycohydrolase (PARG). PAR synthesis involves the cleavage of NAD+ into ADP-ribose (monomers) and free nicotinamide (Virag & Szabo, 2002). A steady-state balance is maintained in normal cells regarding PAR synthesis and degradation. An excellent review on the roots and developments of PARylation research has been published recently (Virag, 2013).

As alterations in PARP or poly-ADP-ribosylation (PARylation) levels are detected in several pathological conditions (Cerboni et al., 2010; Masutani, Nakagama & Sugimura, 2005; Strosznajder et al., 2012; Virag & Szabo, 2002), and PARP or PARG inhibition interferes with T. cruzi infection and proliferation of the parasite (Vilchez Larrea et al., 2012; Vilchez Larrea et al., 2013), PAR biology studies may have far reaching biomedical implications.

PARP gene family includes catalytically inactive members (i.e., ARTD-9 and-13), several members with just mono(ADP-ribosyl)ating (MARylating) activity from which only one has been mapped to submembrane domains (ARTD-8 in focal adhesions) and members with putative (tankyrase-2) or proved enzymatic PARylating activity (Hassa & Hottiger, 2008; Hottiger et al., 2010; Vyas et al., 2013). A different gene family codes membrane-bound or secreted MAR-(or even PAR)-synthesizing enzymes, whose activity is always extracellular: ecto-ADP-ribosyl-transferases (ARTC-1 to 5) (Morrison et al., 2006; Hottiger et al., 2010).

Interestingly, different PARPs may have different PARylating activities. For example, tankyrase-1 (TNKS-1) synthesizes oligomers of an average chain length of 20 units without detectable branching while PARP-1 synthesizes large linear or branched polymers (Hottiger et al., 2010).

A plethora of PARP inhibitors can be used to envisage the involvement of different members of PARP family in specific processes. These compounds display diverse binding and in vitro potencies towards PARP-1 and TNKS-1 (Table 1). Human PARG is expressed in alternative splice variants yielding isoforms that localize to different cell compartments (Bonicalzi et al., 2005; Bonicalzi et al., 2003; Ohashi et al., 2003). Cytoplasmic PARG accounts for most of the PARG activity in cells (Meyer-Ficca et al., 2004). Although most PARG activity would be cytoplasmic and most PARP family members can be detected in the cytoplasm, their role inside the nucleus has been better studied. PARP-1 (the single family member located exclusively in the nucleus), nuclear PARP-2 and -3 compete with histone deacetylases for NAD+ consumption. Poly-ADP-ribosylation of chromatin-associated proteins usually correlates with increased histone acetylation, decreased DNA methylation and low chromatin compaction. Thus, PARylation may modulate gene expression and facilitate the access of DNA repair machinery to damaged sites (Tulin & Spradling, 2003). In fact, PARP-1, the most conserved and best studied PARP, plays a role in the recognition of DNA damage. Nevertheless, PARylation has also been reported in heterochromatic contexts (i.e., X chromosome inactivation) (Burkle & Virag, 2013; Dantzer & Santoro, 2013; Lafon-Hughes et al., 2008).

Table 1 Potency of PARP inhibitors towards PARP-1 and TNKSs.

Binding capacity expressed as ΔTm (°C) according to Wahlberg et al. (2012), in vitro IC50 (correspondent citations in the right-most column). 3-AB, 3-aminobenzamide; OLA, Olaparib; PJ34; XAV, XAV 939. While OLA is considered a selective PARP-1 inhibitor, (which does also target PARP-2, -3 and -4; Narwal, Venkannagari & Lehtio, 2012), PJ34 is a moderate potency inhibitor, displaying PARP-1 preference over TNKS-1, with a lower magnitude order and XAV 939 is relatively selective for TNKS-1 and -2, affecting their activity 169 times more than PARP-1 activity. The inhibitory concentrations in cell culture are always higher than in vitro. For example, 3-AB IC50 in vitro is 5 µM and in vivo, it is commonly used at a 5 mM concentration.

	Δ Tm (°C). Interval	In vitro IC50 (µM)	Citation	
	hPARP-1	TNKS-1	TNKS-2	hPARP-1	TNKS-1	TNKS-2		
3AB	1 to 3.99	<0.99	<0.99	5.400			(Vilchez Larrea et al., 2012)	
OLA	>10	<0.99	<0.99	0.005	1.500		(Vilchez Larrea et al., 2012;
Riffell, Lord & Ashworth, 2012)	
PJ34	7 to 9.99	1 to 3.99	1 to 3.99	0.019	0.570	–	(Lehtio, Chi & Krauss, 2013)	
XAV	1 to 3.99	>10	7 to 9.99	2.200	0.013	0.005	(Lehtio, Chi & Krauss, 2013;
Riffell, Lord & Ashworth, 2012)	

TNKS-1 maps to endoplasmic reticulum, Golgi, secretion vesicles, epithelial lateral membrane or lysosomes (Bottone et al., 2012; Chi & Lodish, 2000; Hsiao & Simth, 2008; Vyas et al., 2013; Yeh et al., 2006). TNKS-1 can also be recruited to the nucleus by TRF1 (telomere repeat binding factor (1) and accompany NuMa (Nuclear/Mitotic apparatus protein) in spindle poles (Hsiao & Simth, 2008). In MDCK (renal epithelial) cells, TNKS-1 is recruited from the cytoplasm to the lateral plasma membrane upon formation of E-cadherin-based cell–cell contacts (Yeh et al., 2006). Extracellular calcium chelation, which prevents cell–cell adhesion, displaces TNKS-1 (Yeh et al., 2006). E-cadherin binds alpha-catenin and vinculin, actin-binding proteins present at the adherence junctions linking actin microfilaments to cadherin. As vinculin and catenin have been recovered as PARylated proteins in co-immunoprecipitation experiments (Gagne et al., 2008; Gagne et al., 2012), we hypothesized that PAR (synthesized by TNKS-1) would be detectable associated to the adherens junctions. It is envisaged that PAR abundance or scarcity could affect the epithelial structure as well as transcendent critical cell signaling pathways, particularly in pathological situations.

In the present work, we have described through immunofluorescence and confocal microscopy, the subcellular localization of PAR in an epithelial monkey kidney cell line (VERO). In fact, we detected PAR associated to the epithelial belt, in a location that has not been previously reported. We have used PARP inhibitors to demonstrate that the immunodetected signal associated to the epithelial belt is PAR and that if PAR synthesis is precluded, actin cytoskeleton as well as cell shape and cell adhesion are affected. Based on these data and previously reported information, TNKS-1 poses as a plausible candidate as the PARP responsible for PAR synthesis in the epithelial belt. However, the participation of other PARPs, particularly TNKS-2, cannot be ruled out. TNKS-1 knockdowns have proven to be unviable since mitosis and cell viability are dramatically affected (Vyas et al., 2013). Therefore, the mechanisms involved in PAR belt formation will be studied in the future by the implementation of other approaches.

Materials and Methods

Cell culture

Cercopithecus aethiops (green monkey) VERO cells (ATCC CCL-81, Faral-Tello et al., 2012) were cultured in MEM (PAA E15-888) supplemented with 10% FBS (PAA A15-151) and 2 mM L-glutamine at 37 °C and 5% CO2. To perform the experiments, cells were seeded in complete media in 24-well plates on 12 mm-diameter coverslips.

Treatments were continuous and carried in duplicates, in parallel with a common (duplicate) control and the correspondent controls without primary antibodies.

Cytoskeleton disruption

Cytochalasin D (GIBCO PHZ 1063; 2 µM and 20 µM) was added 30 min before fixation.

Incubation with PARP inhibitors or a calcium chelator

Cells were incubated with PAR synthesis inhibitors, namely 5 mM 3-aminobenzamide (3-AB, SIGMA A-0788), 80 µM PJ34 (CALBIOCHEM 528150) or 25 µM XAV 939 (abcam 120897), concomitant to seeding. Extracellular calcium deprivation with 3 mM EGTA was also assayed following the same experimental schedule. In all these cases, cells were fixed 5 h after the continuous treatment initiation. Alternatively, the established monolayers were exposed to different PARP inhibitors treatments: 5 mM 3AB (24 h), 80 µM PJ34 (1 h, 5 h, 7 h), 25 µM XAV 939 (12 h, 24 h) or 250 nM Olaparib (JS Research Chemicals Trading; 6 days) and then were subject to fixation.

Immunostaining

Cells were washed in filtered PBS (fPBS, 0.22 µm pore size), fixed in 4% paraformaldehyde (PFA, unless otherwise stated) in fPBS 15 min at 4 °C or in 10% trichloroacetic acid (TCA; see Fig. S1), washed in fPBS, permeabilized in 0.1% Triton-X100 in fPBS, and immersed in blocking buffer (0.2% Tween, 1% BSA in fPBS) for 30 min. An indirect immunostaining procedure was performed. Briefly, cells were incubated with the specific antibodies, namely 1:1,500 rabbit polyclonal anti-PAR (Beckton Dickinson BD551813), 1:1,000 Tulip chicken polyclonal anti-PAR (#1023), 1:1,000 or 1:100 H10 clone mouse monoclonal anti-PAR antibody (Tulip #1020), or 1:100 mouse anti-vinculin (abcam 18058) diluted in blocking buffer for 2 h at 37 °C. After washing in fPBS/T (0.1% Tween), sections were incubated (1 h, RT), with the correspondent anti-antibodies mix (1:500 to 1:250 anti-mouse-Cy3, 1:1,000 anti-rabbit-Alexa 488, 1:500 goat anti-chicken DyLight 488) in blocking buffer for 1 h at RT. When pertinent, 1:150 phalloidin (Molecular Probes R415 or A22283) was included in the mix. After washing in fPBS/T and fPBS, DAPI counterstaining (1.5 µg/mL in fPBS) and a final wash in fPBS, coverslips were mounted in Vectashield (Vector 94010) and sealed with nail polish. Controls without primary antibody were run in parallel to check the specificity of the detected signals. In addition, a control avoiding the permeabilization step was done in order to check if PAR signal was due to the presence of intracellular or extracellular polymer.

Confocal microscopy and image analysis

Single images or image stacks were recorded with an Olympus FV300 with a Plan Apo 60 ×/1.42 NA oil immersion objective or a Leica TCS SP5 II confocal microscope with a Plan Apo 63×/1.4 NA (or a Plan Apo 100×/1.4 NA) oil immersion objective, with or without digital zoom. To assure signal specificity, original images were taken under the same conditions as reference images of cells not labeled with primary antibodies, at the same confocal session. ImageJ free software was used for image processing (including brightness/contrast adjustment and Gaussian blur filtering).

Results and Discussion

Untreated VERO cells harbor different nuclear and peripheral PAR polymers

Poly-ADP-ribose was detected in nuclear and peripheral localizations using the BD anti-PAR antibody. These signals were detected after 10% TCA or 4% PFA fixation. Given that TCA causes protein precipitation, a stronger background was detected in the absence of primary antibody; therefore, PFA was selected for subsequent experiments (Fig. S1).

Since it has recently been demonstrated that at least one member of the ecto-ARTC family can catalyze the synthesis of short lineal PAR chains on the extracellular side of the plasma membrane (Morrison et al., 2006), we decided to check whether the detected epitope was intracellular or extracellular. Hence, immunolocalization was performed avoiding the permeabilization step (in parallel to the routine protocol). In the absence of permeabilization, neither the nuclear nor the peripheral PAR signals were detected (Fig. S2), demonstrating the intracellular nature of the epitope.

Immunostaining with different primary antibodies in parallel yielded apparently conflicting results. For example, nuclear PAR was detected with polyclonal rabbit BD or chicken Tulip anti-PAR antibodies (Fig. 1), but not with Tulip monoclonal H10 clone antibody. According to the respective data sheets, the three antibodies were generated against PAR of unknown length and branching conjugated to methylated BSA. Nevertheless, the latter antibody has known specificity for long PAR chains (above 20 residues; Kawamitsu et al., 1984) and has been widely used to monitor the nuclear response to DNA damage, which is mainly PARP-1 dependent (Vodenicharov et al., 2005; Gagne et al., 2008). PARP-1 synthesizes long branched chains (Hottiger et al., 2010). Coherently, DNA damage response proteins such as p53 or XPA form complexes mainly in the presence of long PAR chains (Fahrer et al., 2007). In fact, while short PAR chains (16-mer) do not interact with XPA and form a single complex with p53, long PAR chains (55-mer) promote the formation of a complex with XPA and three specific complexes with p53 (Fahrer et al., 2007).

Figure 1 PAR pools detection with different anti-PAR antibodies.

PAR (green). Under control conditions, Tulip clone H10 anti-PAR antibody, known to target long ramified PAR, displayed no signal (data not shown). Nuclear PAR was detected both with (A) BD rabbit anti-PAR antibody (#551813) and (B) Tulip chicken anti-PAR antibody (#1023). Peripheral PAR was detected only with BD anti-PAR (A) suggesting differential structures of PAR polymer pools. Bar: 10 µm.

PAR belt was detected with BD rabbit anti-PAR antibody (#551813) but not with Tulip chicken anti-PAR antibody (#1023) (Fig. 1), suggesting again the existence of a differential structure of both PAR polymers. Interestingly, this is not the first report of differential recognition of PAR polymers by antibodies. For example, 16B antibody, which has a preference for branching regions, recognizes just 50% of PAR polymer detectable by H10 (Kawamitsu et al., 1984). Although this phenomenon is more likely to occur with monoclonal antibodies, it seems to be also true for some polyclonal antibodies. In any case, this PAR would correspond to short-chain polymer (up to 20-mer), not recognizable by H10, as expected under the hypothesis that belt PAR is an oligomer (up to 20 units) synthesized by TNKS-1.

Specific PARP inhibition alters microfilament distribution

In VERO cells, PARP-1 is exclusively nuclear (data not shown). The nuclear PARPs inhibitor OLA (250 nM, 6 days; Narwal, Venkannagari & Lehtio, 2012) depleted nuclear PAR without affecting belt PAR (Fig. S3). Treatments with other PARP inhibitors (5 mM 3AB for 24 h, 80 µM PJ34 up to 7 h, 25 µM XAV 939 up to 24 h) applied on the established monolayer had no effect on the PAR belt (data not shown). Taken together, these results suggest that belt PAR is characterized by extremely high stability.

To confirm PAR identity, we evaluated the influence of PARP inhibitors (5 mM 3-AB or 80 µM PJ34) on cellular PAR synthesis. To that end, inhibitors were added at the moment of cell seeding (Fig. 2, green). 3-AB slightly affected PAR while PJ34 showed a stronger effect, particularly on peripheral PAR. The combined treatment yielded a result similar to PJ34 alone. 3-AB is a non potent PARP inhibitor with detectable binding to PARP-1 but undetectable effect on TNKS-1 (Wahlberg et al., 2012; Riffell, Lord & Ashworth, 2012, Table 1). PJ34 binds both nuclear PARPs and TNKS with higher affinity than 3-AB, but the IC50 for PARP-1 is nearly 30 times lower than that reported for TNKS-1. Thus, it affects PARP-1 activity preferentially, but inhibits TNKS-1 and-2 in a non-negligible proportion.

Figure 2 PARP inhibitors diminished PAR belt synthesis.

(A)–(O) VERO cells were fixed 5 h after seeding in the presence of the indicated drugs. (A)–(E) actin (red), (F)–(J) PAR (green), (K)–(O) merge. (A, F, K) control, (B, G, L) 0.5% dimethyl sulfoxide (DMSO; vehicle control), (C, H, M) 5 mM 3-AB, (D, I, N) 80 µM PJ34, (E, J, O) 5 mM 3-AB + 80 µM PJ34. Bar: 25 µm.

In 3-AB and PJ34 experiments, rhodamine-phalloidin (Fig. 2, red) allowed the concomitant detection of the actin cytoskeleton.

3-AB displayed a border-line effect on PAR and actin distribution (Figs. 2C, 2H, 2M). PJ34 induced a deeper blocking effect on PAR belt synthesis (Figs. 2D, 2I, 2N) and the strongest effect was obtained with the combined drugs (Figs. 2E, 2J, 2O). Notice that in spite of diminished belt synthesis, there has been no diminution in cellular density. However, cell shape appeared semi-rounded in some cases and cell area is diminished, reflecting incomplete spread out of the cells. The vehicle, DMSO, affected neither PAR nor the actin belt (Figs. 2B, 2G, 2L).

To sum up, PARP inhibitors affected not only the PAR belt but also the distribution of actin filaments, suggesting the existence of a physical direct or indirect interaction of PAR with the actin cytoskeleton.

Peripheral PAR colocalized with cortical actin and vinculin in the epithelial belt

VERO epithelial cells present adhesion belts separating apical and basal domains, with cortical actin filaments anchored to the belts. To analyze PAR localization in more detail, confocal stacks of cells immunostained for PAR and co-stained with phalloidin (Fig. 3) or co-immunostained to detect vinculin (Fig. 4), were used. Figure 3 highlights the fact that PAR is associated to submembrane domains only in the proximity of a neighbor cell. PAR distribution in the intercellular limits, novel to our knowledge, punctuated lines in intercellular not fully formed contact regions (Figs. 3D–3F, double arrows) and present as a single punctuated line in completely joined cells (Fig. 3C, single arrows) but absent in membrane/cortical domains without neighbor cells (Fig. 3, arrowheads). PAR was located at the place where cortical filaments were anchored, as evidenced by the unequal filament direction between both sides of the intercellular limit/adhesion belt. PAR seemed to be a partner of cortical actin filaments. Z-stacks revealed the existence of a structure that we called the “PAR belt”, with a height of around 1 to 1.5 µm (up to 4 slices every 0.5 µm). The clear-cut presence of PAR in intercellular junctions (arrows) but not associated to the plasma membrane in neighbor-free domains (arrowheads) is illustrated in Fig. 3 orthogonal views (G–R). The yellow lines indicate the cutting planes.

Figure 3 PAR vs. actin in cell–cell adhesions.

(A)–(F) Overview; XY confocal slices (A) actin microfilaments (red), (B) PAR (green), (C) merge + DAPI. Strong PAR signal delineated cell–cell adhesion membrane domains whereas no signal was observed in colony borders. Bar: 10 µm. (D) actin microfilaments (red), (E) PAR (green), (F) merge + DAPI. In immature cell joints, each cell carried its own PAR pool. Thus, two parallel PAR lines were visible. Once membranes joined, a single PAR contour was evident. Bar: 10 µm. (G)–(R) Orthogonal views (XY, XZ, YZ) of a z stack of two neighbor cells. Yellow lines indicate cutting levels. Two main cells and the border of other two cells are visible. (G, M, L, R) XY (z-projection), (H, I, N, O) XZ plane, (J, K, P, Q) YZ plane. (M, N, J, P) actin (red), (I, K, L, O, Q, R) PAR (green), (H) merge, (G) merge + DAPI. Arrows, PAR; double arrows, parallel PARylated cell membranes in an immature cell junction; arrowheads, absent PAR in membranes lacking neighbor cells. Bar: 5 µm.

Figure 4 PAR and vinculin colocalization in the adhesion belts.

Orthogonal views (XY, XZ and YZ) of a z-stack. (A) XY view (z-projection). PAR (green) + vinculin (red) + DAPI (blue). (B) XZ view. (C) YZ view without DAPI; (D) XY view. PAR (green). Arrows, PAR + vinculin in the PAR belt; arrowheads, non-PARylated vinculin in cell-matrix junctions. Bar: 5 µm.

Vinculin is an actin-binding protein that displays a dual localization: basal and apical, related to cell-matrix focal adhesions and to ZO-1 positive tight junctions in the epithelial belt, respectively (Maddugoda et al., 2007). Interestingly, while focal adhesion vinculin is not PARylated (Fig. 4, arrowheads), a colocalization of PAR (in green) and vinculin (in red) is observed at the apical position correspondent to the epithelial belt (Fig. 4, arrows).

During actin cytoskeleton disruption, PAR accompanied actin

In order to test the physical association of PAR to the actin cytoskeleton in this particular localization, we induced microfilament disassembly through cytochalasin D (2 and 20 µM, 30 min) treatment. While in control cells the actin/PAR belt could clearly be observed (Figs. 5A–5D, arrow), after 30 min of low-dose cytochalasin D treatment (2 µM) the actin belt was fractured. In some cells, belt microfilaments could still be observed (Figs. 5E–5H, arrows), although they looked non-continuous and tensionless; in other cells, the actin/PAR belt had disappeared (Figs. 5E–5H, arrowheads). After exposure to a stronger treatment (20 µM cytochalasin D), no filamentous structure remained. Instead, there was a punctuated actin pattern with some precipitates (Figs. 5I–5L, arrow) colocalizing with PAR. Therefore, belt PAR accompanied actin microfilaments during their structural loss.

Figure 5 Cytochalasin D induced PAR delocalization together with actin depolymerization.

(A)–(D) control, (E)–(H) 2 µM cytochalasin, (D, I, L) 20 µM cytochalasin D. (A, E, I) actin (red), (B, F, J) PAR (green), (C, G, K) DAPI (blue), (D, H, L): merge. Arrows, PAR coexisting with actin; arrowheads, PAR belt absence where actin is absent. Bar: 10 µm.

EGTA or XAV 939 disturbed PAR belt synthesis, affecting the actin cytoskeleton, cell shape and cell adhesion

We reasoned that in a condition in which TNKS-1 was not recruited to the epithelial belt, peripheral PAR would not be synthesized. It is well established that extracellular Ca2+ chelation hampers cell adhesion. More recently, it has been shown that TNKS-1 is recruited from the cytoplasm to the lateral plasma membrane upon formation of E-cadherin-based cell–cell contacts in renal epithelial cells, and the recruitment depends on extracellular calcium ion (Yeh et al., 2006). Thus, we depleted extracellular calcium with EGTA (3 mM). Under this condition, cell roundness and diminished cell adhesion leading to reduced and irregular cell density were observed (although not reflected in the photographs because empty fields were not photographed). Concomitantly with cell roundedness, PAR diminution was observed (Figs. 6B, 6E, 6H), as expected under our hypothesis.

Figure 6 EGTA and XAV 939 affected the actin cytoskeleton, cell shape and cell adhesion.

(A)–(C) Actin (red), (D)–(F) PAR (green), (G)–(I) merge. (A, D, G) control, (B, E, H) 3 mM EGTA, (C, F, I) 25 µM XAV 939. Bar: 25 µm.

EGTA chelation is a very unspecific treatment. Thus, we next exposed cells from the moment of seeding to XAV 939, an inhibitor which exhibits a strong preference for TNKSs over other PARPs (Table 1; Wahlberg et al., 2012; Riffell, Lord & Ashworth, 2012). Like with EGTA, a decrease in cell density was repeatedly observed (Figs. 6C, 6F, 6I). As the time interval was short (just 5 h), this cannot be explained by a reduction in the number of cell cycles, but by diminished cell attachment. This effect (Figs. 6C, 6F, 6I) is stronger than the one observed with PJ34 + 3AB (Figs. 2E, 2J, 2O) which had reduced cell spreading area without lessening the number of monolayer-constituting cells. There were a plethora of cell shapes including round and binucleated cells. Finally, while in control populations it was difficult to find an isolated cell pair (a confluent monolayer was almost everywhere), in XAV-treated populations cell pairs were frequent, but many times the PAR belt junction was incomplete. To sum up, XAV 939 displayed a strong effect on the cell junction regions, with diminished cell attachment, increased roundness and partial loss of PAR/actin belt.

Conclusions

In the present work we have shown for the first time the existence of a PAR belt associated to the actin cytoskeleton and colocalizing with the anchorage protein vinculin. Vinculin associates to the E-cadherin complex. Thus, it is expected that PAR interacts with several members of the complex. Although our data fits the reported vinculin/alpha-catenin co-immunoprecipitation with anti-PAR antibodies (Gagne et al., 2008; Gagne et al., 2012), in our system, given the resolution of confocal microscopy, we did demonstrate that the cell junction apparatus (not necessarily nor exclusively vinculin) is PARylated.

Actin cytoskeleton disruption affects the PAR belt whereas the interference with PAR belt synthesis leads to actin cytoskeleton, cell shape and cell adhesion changes.

The graded effects obtained with 3AB, PJ34 and XAV 939 indicate a probable involvement of TNKSs in PAR belt synthesis. We cannot formally exclude TNKS-2 as the responsible enzyme. Parsimony favors the hypothesis of TNKS-1 involvement in PAR belt synthesis, since (1) In VERO cells, belt PAR is not detected with an antibody targeting long PAR chains (TNKS-1 synthesizes short chains); (2) in another renal epithelial cell line (MDCK), TNKS-1 is localized at the epithelial belt and its activity is required for cell–cell adhesion (Yeh et al., 2006). Demonstrating that TNKS-1 is responsible for the observed PARylation is a difficult task. The generation of a mammalian cell TNKS-1 knockdown has been attempted (Vyas et al., 2013), but it resulted in an unviable cell line, affecting the whole cell and rapidly leading to cell death, precluding a clear dissection of the underlying mechanisms. Though TNKS-2 participation cannot be ruled out, it should be pointed out that its functionality as a PARylating enzyme has not been proven up to date. The evaluation of the detailed mechanisms involved in PAR belt synthesis as well as the identification of the individual components of the E-cadherin/actin complex which are PARylated, will be the subject of future research. Further work will also be necessary to analyze the existence of the PAR belt in other epithelial cells to fully characterize the biochemical differences among nuclear and belt PAR, and to study the functional implications of the PAR belt in different systems.

Supplemental Information

Figure S1 PAR belt detection in trichloroacetic acid (TCA) - or 4% PFA -fixed cells

Merged DAPI (blue) and PAR (green) channels. (A, B) TCA fixation in the absence (A) or presence (B) of the primary antibody. (C, D) 4% PFA fixation in the absence (C) or presence (D) of the primary antibody. All the photographs were taken on the same confocal session under the same conditions and were equally processed. PAR belt signal is clear in both cases, but the background is lower with 4% PFA.

Click here for additional data file.

Figure S2 PAR belt is intracellular

(A)–(C) PAR (green), (D)–(F) merged PAR (green), actin (red) and DAPI (blue). (A, D) Control (usual protocol), (B, E) same protocol except for the absence of permeabilization, (C, F) control with permeabilization without primary antibody. Bar: 10 µm.

Click here for additional data file.

Figure S3 Olaparib depleted nuclear PAR without affecting peripheral PAR in Vero cells

PAR (green). (A) control, (B) Olaparib (250 nM, 6 days). Bar: 20 µm.

Click here for additional data file.

We are indebted to MSc Pablo Liddle, the technician from the Confocal Microscopy Service, Facultad de Medicina, Universidad de la República, who assisted us with photography through LEICA confocal microscope. We are also indebted to Gustavo Folle, Maria Vittoria Di Tomaso and Ana Laura Reyes for stimulating discussions. Finally, we are grateful to Santiago Mirazzo and Juan Arbiza for the cell lines.

Additional Information and Declarations

Competing Interests

Author Contributions

Data Deposition

The authors declare there are no significant competing financial, professional or personal interests that might have influenced the performance or presentation of the work described in this manuscript.

Laura Lafon-Hughes conceived and designed the experiments, performed the experiments, analyzed the data, contributed reagents/materials/analysis tools, wrote the paper, prepared figures and/or tables, reviewed drafts of the paper.

Salomé C. Vilchez Larrea, Alejandra Kun and Silvia H. Fernández Villamil conceived and designed the experiments, analyzed the data, contributed reagents/materials/analysis tools, reviewed drafts of the paper.

The following information was supplied regarding the deposition of related data:

https://drive.google.com/a/peerj.com/folderview?id=0B3UddQSszJZqS3c3alo5a1NwS1k&usp=sharing.

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
