# Peer review of "VERO cells harbor a poly-ADP-ribose belt partnering their epithelial adhesion belt"

_PeerJ, doi:10.7717/peerj.617_

## Round 0.1 · original submission · Minor Revisions

The present manuscript has some new informationson the novel roles of poly-ADP-ribose (PAR) in the cytoskeletal potentials. The logic explanation how TNKS-1 displays for the PAR function in the cytoskeletal arrangements should be discussed in the appropriate section during your revision. We understand this study is not the final destination of the TNKS-1-PAR-cytoskeletal collaborative expression.

Reviewer 1 ·

Basic reporting

No comments

Experimental design

No comments

Validity of the findings

No comments

Additional comments

In the paper, the authors suggest the existence of PAR in a novel subcellular localization.
The paper may be acceptable for publication provided major corrections are made and additional information is given. Please refer to the details below.
Major issues:
1-How is PARP and TNKS-1 expressed in VERO cells? The authors only show PAR accumulation, but they must also show colocalisation of PAR-TNKS-1, and PAR-PARP1 on VERA cells. And the authors must confirm the result with western blot analysis for PARylated actin, vinculin, TNKS-1, PARP1.

2-In figure 6 the TNKS-1 inhibitor XAV939 does not completely inhibit the “PARP epithelial adhesion belt”. Given the high potency of XAV939, this result does not seem to agree with the author’s hypothesis. The authors should perform additional experiments in which XAV939 is added only after seeding of the cells to see if the PAR belt disappears later.

3-The authors showed that “Figure 3 highlights the fact that PAR is associated to sub membrane domains only in the proximity of a neighbor cell.” But PAR staining in figure 2 (F) and in figure 3 (B) look different. They used same antibody for the same specimen, the authors must explain why PAR staining is different?

Minor issues:
1-The authors should explain more clearly why they choose 3-AB, Olaparib, PJ34, XAV 939.

2-In material and methods, it is written “In all cases, cells were fixed 5 h after treatment initiation.” Is this means all treatment (PARP inhibitors and EGTA) were for 5 hours? In the “Results and Discussion” part it is specified “accordingly, nuclear PARPs inhibitor (Narwal et al. 2012) Olaparib (250nM, 6 days)”. Is the treatment with Olaparib for 6 days? It is confusing and the authors must explain the protocol more clearly in the material and method part.

3-In figure 2, treatment with 3AB and PJ34+3AB showed differences in actin structures, authors must discuss and explain this in the results and discussion part. According to Riffel et al, 2012, PJ34 exhibits at least 30X higher potency towards PARP1 than to TNKS-1. Why then does PJ34 treatment result suggest TNKS-1 involvement?

4-Many panels of Figure 2 are not indicated in the text. The authors should give detail to which panel exactly they are referring to.

5-In the introduction part, the reference style (Virag 2013) is not similar like others. The authors should write it in similar way.

Reviewer 2 ·

Basic reporting

No Comments

Experimental design

No Comments

Validity of the findings

No Comments

Additional comments

In the manuscript, Laura Lafon-Hughes et al. have investigated the subcellular localization of PAR in an epithelial monkey kidney cell line (VERO) and demonstrated that the existence of PAR in a novel subcellular localization and consistented with the view that such PAR may be synthesized by TNKS-1 . This work could be of interest in Poly-ADP-ribose (PAR) research area. However, some important concerns should be addressed:

In the manuscript, the subcellular localization of PAR in an epithelial monkey kidney cell line (VERO) and demonstrated that the existence of PAR in a novel subcellular localization. So, the pictures for the results are very important. But in the “Results”, legend, the FIGURE LEGENDS of pcture 4 are not matching with the pictures 4, especial for C and D. Please check them and correct them.

When suitably revised, this paper will make a very interesting contribution to the growing literature on the Poly-ADP-ribose (PAR) research area.

Reviewer 3 ·

Basic reporting

The manuscript requires editing in some areas. For example:

Line 210: "went along" is not scientific language
Line 231: IC50 values for TNKS-1 not shown

Also, the authors need to expand on descriptions and explanations of experimental findings in the results section, as well as the conclusions. These are important edits that can easily be accomplished upon revision.

Experimental design

The authors present very impressive visual images of their findings. However, more clarity is required for some figures:

Fig. S3: was there really 6 days of olaparib treatment?
Fig. 2: what was the total length of time for PARP inhibitor treatments?
Fig. 4: confusing--is red staining for actin or vinculin? If actin, how was vinculin detected and visualized?
Fig. 5: This is impressive data, with striking images. Please expand explanations in lines 211-214. What do the arrows and arrowheads indicate? What can be concluded from this figure?

Validity of the findings

The primary concern is that XAV939 also inhibits tankyrase-2 and the authors repeatedly implicate tankyrase-1 in the findings. The authors need to clearly acknowledge that TNKS-2 may be a possibility in the results and conclusion sections.

Also, a minor point is that it is well-known that different PAR antibodies detect different types of PAR. Most were engineered to detect complex and long PAR that results from DNA damage. Since all the anti-PAR antibodies used were polyclonal, it may help the explanation and justification of the results by discussing how each antibody was raised. Via BSA-hapten? With long or short PAR?

Additional comments

This manuscript presents very impressive visual images that uncover novel roles of poly-ADP-ribose (PAR) in the mediation of cytoskeletal functions. However, the hypothesis that TNKS-1 is the enzyme responsible for this is a bit too premature. Also, the justification for the use of a particular fixation technique and a particular PAR antibody is a bit confusing as well. But the data is unique and does allow for new contributions to the fields of PAR and cytoskeletal dynamics. With minor revisions in text and editing, i.e. better explanations and justifications, acknowledgement that TNKS-1 may not be the primary enzyme responsible, this manuscript will be suitable for publication in PeerJ.

Reviewer 4 ·

Basic reporting

The manuscript meets the criteria of basic reporting. The chapters are well structured and can be followed logically as well. Relevant prior literature is appropriately referenced.
Minor revisions for the authors:
-Fig. 5, 6 is not mentioned in the text. For a proper conclusion they should be featured.
-The abbreviation of "PFA" is not introduced in the text.
-Supplier, catalog number and animal species is missing for H10 clone anti-PAR antibody.
-anti-chicken anti-antibody is not mentioned on the list of secondary antibodies.
-The type of antibodies (mono-, or polyclonal) should be stated because this could be linked to the different recognition properties of the used antibodies
-The paragraph starting from line 148 in the Results section is ambiguous. Avoiding the permeabilization step allows to study only extracellular antigens not intracellular. Therefore I suggest changing the word intracellular to extracellular in the sentence: ...we decided to check the intracellular nature of... For this reason the last sentence should also be deleted because it states an evidence in this context.
-In line 174 the authors mention Olaparib treatment (250nM, 6days) which also leads to misunderstandings. In the methods section only 5 hours treatment is mentioned. This should be clarified.
-Such comparison of IC50 values in line 174 and line 181 is not widespread in the literature. Nevertheless, it also leads to a wrong conclusion. For Oliparib the authors' comparison claims that it is a more potent inhibitor on TNKS-1 and for PJ34 that it is more potent on PARP-1. I suggest using the exact IC50 values for comparing the affect of the drugs on the enzymes.
-Using a sub-heading before the paragraph that starts from line 176 could make the whole text more transparent. The results discussed from here are coming from treated VERO cells but the chapter start from "Untreated Vero cells"...
-Figure 5 legend should be changed as follows: ...cytochalasin D, (I-L)...
-Figure 4: which color is for vinculin?

Experimental design

The research described in the manuscript is relevant and meaningful. The experimental design meets the technical standards and the methods are described with sufficient information.
Minor revisions for the authors:
-Using a positive control for long branched PARylation (e.g.: oxidative stress for the cells) would clarify the negative results for H10 antibody. The authors properly explain this phenomenon with the preference of H10 antibody for long PAR chains but an experimental observation would undoubtedly demonstrate the validity of that assumption.

Validity of the findings

The findings of the manuscript are very interesting and worth publishing. The conclusions about the existence of a "PAR belt" are appropriately stated and limited to those supported by the results. However these observations should also be supported by independent methods. For example a fractionation experiment would clarify the specificity of the antibodies. Proximity ligation could also confirm the existence of a peripheral, actin associated PARylation process. An immune-independent method would be the use of tritium labeled NAD for tracking compartment specific PARP activity.

---

## Round 0.2 · accepted · Accept

Your revision is now acceptable for publication and we appreciate you for the careful address and correction.

---

## Author Rebuttal · Round 0.2

Thank you for your submission to PeerJ. I am writing to inform you that in my opinion as the Academic Editor for your article, your manuscript "VERO cells harbor a poly-ADP-ribose belt partnering their epithelial adhesion belt" (#2014:06:2240:0:2:REVIEW) requires some minor revisions before we could accept it for publication.

The comments supplied by the reviewers on this revision are pasted below. My comments are as follows:

## Editor's comments

The present manuscript has some new information on the novel roles of poly-ADP-ribose (PAR) in the cytoskeletal potentials. The logic explanation how TNKS-1 displays for the PAR function in the cytoskeletal arrangements should be discussed in the appropriate section during your revision. We understand this study is not the final destination of the TNKS-1-PAR-cytoskeletal collaborative expression.

Cheorl-Ho Kim
Academic Editor for PeerJ

Dear Editor and Reviewers,

We are currently submitting the revised manuscript version. We have considered all your comments. As you stated, the objective of this work is just to demonstrate the existence of the PAR belt. TNKS-1 subcellular localization and PARilated vinculin immunoprecipitation done in other cellular systems inspired this work. In the present work, we do not aim to demonstrate that PAR is synthethized by TNKS-1, references to this PARP only intend to point out that the experimental data obtained could argue in this direction. This will be discussed more thoroughly in the manuscript, to avoid this confusion.

# Reviewer Comments

## Reviewer 1

**Basic reporting**

No comments

**Experimental design**

No comments

**Validity of the findings**

No comments

**Comments for the author**

In the paper, the authors suggest the existence of PAR in a novel subcellular localization. The paper may be acceptable for publication provided major corrections are made and additional information is given. Please refer to the details below.

Major issues:

1-How is PARP and TNKS-1 expressed in VERO cells? The authors only show PAR accumulation, but they must also show colocalisation of PAR-TNKS-1, and PAR-PARP1 on VERA cells. And the authors must confirm the result with western blot analysis for PARylated actin, vinculin, TNKS-1, PARP1.

As suggested by reviewer 1, we have studied the subcellular localization of PARP-1 in Vero cells. As can be seen in Figure I, detectable PARP-1 is exclusively nuclear, arguing against PARP-1 dependent PAR belt synthesis. Our first aim is to communicate the presence of the observed PAR adhesion belt, and although we considered the observation to be very pertinent, we are gathering the necessary tools to work thoroughly on the relationship between PAR and TNK in the near future.

[Figure]

Figure I. PARP-1 detection in VERO cells with rabbit anti-PARP antibody (Santa Cruz sc-7150, 1:300)

We have performed the suggested western blot of untreated (control) cells (Figure II). The pattern detected with anti-vinculin antibody is exactly like the one observed in the fabricator´s datasheet. The stronger  band scarcely below 130 Kda corresponds to whole-length vinculin. The 90 kDa band might correspond to the head domain of cleaved vinculin.

[Figure] [Figure]

Figure II.- PARylated proteins detected in VERO cells with BD anti-PAR antibody in control condition. (A) western blot from VERO whole cell  extract to detect vinculin and PAR (pADPr). (B) Overview of immunolocalization of vinculin (red) and PAR (green). While vinculin is found throughout the cytoplasm, PAR is only at the tiny epithelial belt.

There is a strong signal of PARylated cell components of unknown identity around 60 kDa (non-correspondent to actin, vinculin, TNKS-1 or PARP1). A weaker signal corresponding to proteins in the 70-170 kDa molecular range can also be observed. Besides, PARylation can be covalent or non-covalent, and the latter could be lost under denaturing conditions. Therefore, we cannot assure whether a vinculin fraction is PARylated or not. The results are highly inconclusive. Our future work is likely to include cell fractionation experiments, immunoprecipitation and mass spectrometry protein identification of PARylated proteins in VERO cells.

2-In figure 6 the TNKS-1 inhibitor XAV939 does not completely inhibit the "PARP epithelial adhesion belt". Given the high potency of XAV939, this result does not seem to agree with the author's hypothesis.

We believe that the incomplete inhibition of the PAR epithelial adhesion belt can be very likely due to permeability problems. Although XAV939 has been demonstrated to have a good cell-permeability level, the stronger inhibitory effect of XAV939 after cell electroporation suggests that the internalization of the inhibitor may be slow, arguing in favor of this hypothesis.

[Figure]

Figure III. Effect of XAV939 (50µM) on electroporated cells. XAV939 was added immediately after electroporation (during incubation in ice). Then cells were seeded. Media was changed daily to renew XAV939 (or DMSO alone . Cells were fixed three days later. A-D electroporated cells; E-H electroporated cells + 50 uM XAV939. Notice the deficient cell adhesion and the lack of a PAR / actin belt. XAV-treated cells display apoptotic features, including nuclear PAR increase and chromatin condensation.

The authors should perform additional experiments in which XAV939 is added only after seeding of the cells to see if the PAR belt disappears later.

We have performed the suggested experiments using 5mM 3-AB 24 h), 100 nM EB (24 h), 80 µM PJ34 (1h and 7h) or 25 µM XAV939 (12 h or 24 h). Nevertheless, we could not induce belt disassembly. Once established, the belt is very stable.

3-The authors showed that "Figure 3 highlights the fact that PAR is associated to sub membrane domains only in the proximity of a neighbor cell." But PAR staining in figure 2 (F) and in figure 3 (B) look different. They used same antibody for the same specimen, the authors must explain why PAR staining is different?

The differences pointed by the reviewer regarding PAR staining in figures 2 and 3 are due to differences in the brightness of the image. As it can be observed when comparing controls, in figure 2 the brightness is higher than in figure 3, but the staining pattern is conserved and is consistent between the images. Within each experiment, all images from different treatments are processed in exactly the same way.  But image processing parameters can be changed  from one experiment to another. If we diminish brightness from all images in Figure 2 (so that the control will be equally bright to the control in Figure 3B), it will be impossible to see PAR in Figure 2J. We tried to display the results in the most conservative way. Raw data has been provided to the journal and can be accessed to corroborate these statements.

Minor issues:
1-The authors should explain more clearly why they choose 3-AB, Olaparib, PJ34, XAV 939.

Inhibitors used were chosen on their differential binding/inhibiting capacities and their availability. 3-AB is a broad range PARP inhibitor, and though it displays low potency, it is readily available from several companies in South America. Less common, Olaparib displays  stronger potency towards nuclear PARPs (1 and 2) and  inhibits PARP-3 as well. It is 300 times more potent as a PARP-1 inhibitor than as a TNKS-1 inhibitor. PJ34 is just 30 times more potent as a PARP-1 inhibitor than as a TNKS-1 inhibitor. Thus, any effect on TNKS-1 will be observed more probably using PJ34 than OLA. On the other hand, XAV939 has high binding potency towards TNKS and low affinity for PARP-1, -2, and -3. Moreover, it inhibits TNKS with a potency that is 169 times that for PARP-1. A graph and a table with the corresponding information is provided below and the properties that led us to choose the inhibitors used was addressed in the manuscript for further clarity.

| | Δ Tm (ºC). Interval. | | | In vitro IC50 (µM) | | | Citation |
|---|---|---|---|---|---|---|---|
| | hPARP-1 | TNKS1 | TNKS2 | hPARP-1 | TNKS1 | TNKS2 | |
| 3AB | 1 to 3.99 | < 0.99 | < 0.99 | 5.400 | | | Vilchez et al. 2012 |
| OLA | > 10 | < 0.99 | < 0.99 | 0.005 | 1.500 | | Vilchez et al. 2012, Riffell et al. 2012 |
| PJ34 | 7 to 9.99 | 1 to 3.99 | 1 to 3.99 | 0.019 | 0.570 | - | Lehtio et al. 2013 |
| XAV939 | 1 to 3.99 | > 10 | 7 to 9.99 | 2.200 | 0.013 | 0.005 | Lehtio et al. 2013, Riffell et al. 2012 |

2-In material and methods, it is written "In all cases, cells were fixed 5 h after treatment initiation." Is this means all treatment (PARP inhibitors and EGTA) were for 5 hours? In the "Results and Discussion" part it is specified "accordingly, nuclear PARPs inhibitor (Narwal et al. 2012) Olaparib (250nM, 6 days)". Is the treatment with Olaparib for 6 days? It is confusing and the authors must explain the protocol more clearly in the material and method part.

The generalization applied to all the experiments has been the one included in the manuscript body. The experiment with Olaparib (250nM) follows a different time schedule (6 days) which was not included in Material and Methods because the result is displayed as a supplemental figure. Nevertheless, as this induced confusion, we have now included this schedule under Material and Methods.

3-In figure 2, treatment with 3AB and PJ34+3AB showed differences in actin structures, authors must discuss and explain this in the results and discussion part. According to Riffel et al, 2012, PJ34 exhibits at least 30X higher potency towards PARP1 than to TNKS-1. Why then does PJ34 treatment result suggest TNKS-1 involvement?

We understand that the way of expressing this idea was not adequate; it induced to confusion. Therefore, we have changed the drafting (the way of writing) to avoid confusion:

*PJ34 binds both nuclear PARPs and TNKS with higher affinity than 3-AB, but the IC50 for PARP-1 is nearly 30 times lower than that reported for TNKS-1. Thus, it affects PARP-1 activity preferentially, but inhibits TNKS-1 and-2 in a non-negligible proportion.*

The results with PJ34 prompted us to buy XAV 939. We still consider that, even after using XAV939, we have not demonstrated TNKS-1 involvement; we are just showing that there are elements pointing logically to this hypothesis.

4-Many panels of Figure 2 are not indicated in the text. The authors should give detail to which panel exactly they are referring to.

This has been corrected.

5-In the introduction part, the reference style (Virag 2013) is not similar like others. The authors should write it in similar way.

The reason why this reference does not look like others, is due to that is paper with a single author, but the style is the same as the rest of the references in the manuscript, since ENDNote was used for format all references.

## Reviewer 2

**Basic reporting**

No Comments

**Experimental design**

No Comments

**Validity of the findings**

No Comments

**Comments for the author**

In the manuscript, Laura Lafon-Hughes et al. have investigated the subcellular localization of PAR in an epithelial monkey kidney cell line (VERO) and demonstrated that the existence of PAR in a novel subcellular localization and consistented with the view that such PAR may be synthesized by TNKS-1 . This work could be of interest in Poly-ADP-ribose (PAR) research area. However, some important concerns should be addressed:

In the manuscript, the subcellular localization of PAR in an epithelial monkey kidney cell line (VERO) and demonstrated that the existence of PAR in a novel subcellular localization. So, the pictures for the results are very important. But in the "Results", legend, the FIGURE LEGENDS of picture 4 are not matching with the pictures 4, especial for C and D. Please check them and correct them.

It has been corrected in the manuscript.

When suitably revised, this paper will make a very interesting contribution to the growing literature on the Poly-ADP-ribose (PAR) research area.

## Reviewer 3

**Basic reporting**

The manuscript requires editing in some areas. For example:

Line 210: "went along" is not scientific language

It has been substituted by "accompanied"

Line 231: IC50 values for TNKS-1 not shown

We agree the notation previously used could be confusing. The IC50 of XAV 939 is 2.2 µM for PARP-1 and 0.011-0.013 nM for TNKS1 (Riffell et al. 2012, Lehtio et al. 2013). Therefore, the inhibitory potency of XAV939 is 200 times higher for TNKS1 than for PARP-1. That´s what we meant when we wrote "IC50 PARP-1= 200 IC50 TNKS-1". Now we have included this explanation in the manuscript.

Also, the authors need to expand on descriptions and explanations of experimental findings in the results section, as well as the conclusions. These are important edits that can easily be accomplished upon revision.

**Experimental design**

The authors present very impressive visual images of their findings. However, more clarity is required for some figures:

Fig. S3: was there really 6 days of Olaparib treatment?

As indicated above for the observation of reviewer 1, the experimental conditions have clearly been thoroughly described in the Materials and Methods section.

Fig. 2: what was the total length of time for PARP inhibitor treatments?

The total length of time for PARP inhibitor treatments was 5 h (except on de OLA experiment where it was six days).

Fig. 4: confusing--is red staining for actin or vinculin? If actin, how was vinculin detected and visualized?

In Figure 4, red is vinculin detected with anti-vinculin antibodies, as stated in Materials and Methods section. Vinculin staining pattern is different from actin pattern. Vinculin is present mainly at focal adhesions, participating in cell-substrate junctions and more apically, in the adhesion belt. It is also observed as a punctuated instead of filamentous image. The Figure legend has been corrected accordingly (where it read actin we meant  vinculin).

Fig. 5: This is impressive data, with striking images. Please expand explanations in lines 211-214. What do the arrows and arrowheads indicate? What can be concluded from this figure?

As stated in the figure legend, arrows point PAR, which colocalizes with actin whereas arrowheads show that the PAR belt disappears concomitantly with the actin belt. This explanation has now been included in the main text.

**Validity of the findings**

The primary concern is that XAV939 also inhibits tankyrase-2 and the authors repeatedly implicate tankyrase-1 in the findings. The authors need to clearly acknowledge that TNKS-2 may be a possibility in the results and conclusion sections.

The observation is pertinent. It is true that XAV939 inhibits TNKS-2 with a potency similar or even slightly higher than TNKS1. Now, we have explicitly stated the formal possibility that TNKS-2 is involved. Our initial consideration for TNKS-1 relayed mostly on the fact that literature for TNKS subcellular localization as well as reports on the role of TNKS-1 in cell-cell adhesion and Wnt signaling, points towards TNKS-1 as the most probable actor in this phenomena.

Also, a minor point is that it is well-known that different PAR antibodies detect different types of PAR. Most were engineered to detect complex and long PAR that results from DNA damage. Since all the anti-PAR antibodies used were polyclonal, it may help the explanation and justification of the results by discussing how each antibody was raised. Via BSA-hapten? With long or short PAR?

Tulip 1020 (clone H10) is mouse monoclonal antibody whereas BD 551813 is rabbit polyclonal antibody and Tulip 1023 is chicken polyclonal antibody. According to the datasheets, the three of them were generated against purified poly(ADP-ribose) of unknown branching or length mixed with methylated BSA.

As stated in the text, Fahrer et al. (2010) demonstrated that the interaction among PAR and particular proteins depends on PAR chain length. Besides, two antibodies generated against PAR have different capacity to recognize lineal vs branched polymer (Kawamitsu et al. 1984).

**Comments for the author**

This manuscript presents very impressive visual images that uncover novel roles of poly-ADP-ribose (PAR) in the mediation of cytoskeletal functions. However, the hypothesis that TNKS-1 is the enzyme responsible for this is a bit too premature.

Also, the justification for the use of a particular fixation technique and a particular PAR antibody is a bit confusing as well.

We have initially used two fixation techniques: 4% PFA and 10 %TCA. Then, we continued the work using the one which preserves better the 3D cell structure.

But the data is unique and does allow for new contributions to the fields of PAR and cytoskeletal dynamics. With minor revisions in text and editing, i.e. better explanations and justifications, acknowledgement that TNKS-1 may not be the primary enzyme responsible, this manuscript will be suitable for publication in PeerJ.

We agree with the observation that a conclusion based solely on the results here presented about the responsibility of TNKS-1 in PARP-belt synthesis would be premature and was not our intention to present it as such. We considered valid, however, to present a hypothesis on the subject, as we believe this allows to discuss our results in a larger and deeper context.

# Reviewer 4

## Basic reporting

The manuscript meets the criteria of basic reporting. The chapters are well structured and can be followed logically as well. Relevant prior literature is appropriately referenced.

Minor revisions for the authors:

-Fig. 5, 6 is not mentioned in the text. For a proper conclusion they should be featured.

We have named the figures in the text.

-The abbreviation of "PFA" is not introduced in the text.

We have introduced the abbreviation.

-Supplier, catalog number and animal species is missing for H10 clone anti-PAR antibody.

Now, it has been included: Tulip #1020, mouse monoclonal antibody

-anti-chicken anti-antibody is not mentioned on the list of secondary antibodies.

Now we have mentioned it: anti-chicken-DyLight 488 (abcam 96947, 1:500)

-The type of antibodies (mono-, or polyclonal) should be stated because this could be linked to the different recognition properties of the used antibodies

We have introduced this information in the manuscript.

-The paragraph starting from line 148 in the Results section is ambiguous. Avoiding the permeabilization step allows to study only extracellular antigens not intracellular. Therefore I suggest changing the word intracellular to extracellular in the sentence: ...we decided to check the intracellular nature of... For this reason the last sentence should also be deleted because it states an evidence in this context.

The paragraph was rewritten in order to clarify the information, as follows::

.."we decided to check whether the detected epitope was intracellular or extracellular. To this end, immunolocalization was performed avoiding the permeabilization step (in parallel to the routine protocol). In the absence of permeabilization, neither the nuclear nor the peripheral PAR signals were detected, demonstrating the intracellular nature of the epitope".

-In line 174 the authors mention Olaparib treatment (250nM, 6days) which also leads to misunderstandings. In the methods section only 5 hours treatment is mentioned. This should be clarified.

The generalization applied to all the experiments included in the manuscript body. The experiment with Olaparib (250nM) follows a different time schedule (6 days) which was not included in Material and

Methods because the result is displayed as a supplementary figure. Nevertheless, as this induced confusion, now we have explicitly included this schedule under Material and Methods.

Such comparison of IC50 values in line 174 and line 181 is not widespread in the literature. Nevertheless, it also leads to a wrong conclusion. For Oliparib the authors' comparison claims that it is a more potent inhibitor on TNKS-1 and for PJ34 that it is more potent on PARP-1. I suggest using the exact IC50 values for comparing the effect of the drugs on the enzymes.

The notation that we used was misleading, so it was modified.

-Using a sub-heading before the paragraph that starts from line 176 could make the whole text more transparent. The results discussed from here are coming from treated VERO cells but the chapter start from "Untreated Vero cells"...

We agree. We have added the subheading: Peripheral PAR colocalized with cortical actin and vinculin in the epithelial belt.

-Figure 5 legend should be changed as follows: ...cytochalasin D, (I-L)...

This typing error has been corrected.

-Figure 4: which color is for vinculin?

The reference was  incorrect; we have corrected it indicating that red staining corresponds to vinculin.

**Experimental design**

The research described in the manuscript is relevant and meaningful. The experimental design meets the technical standards and the methods are described with sufficient information.

Minor revisions for the authors:

-Using a positive control for long branched PARylation (e.g.: oxidative stress for the cells) would clarify the negative results for H10 antibody. The authors properly explain this phenomenon with the preference of H10 antibody for long PAR chains but an experimental observation would undoubtedly demonstrate the validity of that assumption.

We have exposed VERO cells to 250 uM $H_2O_2$ for 2 h. H10 clon anti-PAR antibody positive signals were only found in DAPI-positive regions, reinforcing the idea that the clon H10 anti-PAR antibody does detect long branched PAR chains

[Figure]

Figure IV. Long-chain PAR detection with clon H10 anti-PAR antibody in some damaged cells.

**Validity of the findings**

The findings of the manuscript are very interesting and worth publishing. The conclusions about the existence of a "PAR belt" are appropriately stated and limited to those supported by the results. However these observations should also be supported by independent methods. For example a fractionation experiment would clarify the specificity of the antibodies. Proximity ligation could also confirm the existence of a peripheral, actin associated PARylation process. An immune-independent method would be the use of tritium labeled NAD for tracking compartment specific PARP activity.

In addition to the immunodetection of PARP, we have performed and experiment based on the incorporation of biotynilated NAD$^+$ in electroporated cells, subsequently detected with fluorescent streptavidin (Figure V).

For this experiment, we first analyzed the sucellular localization of endogenous biotin in VERO cells: it is widespread in nuclei and cytoplasm but absent in the plasma membrane.

[Figure]

Figure V. Cells electroporated with biotinylated NAD$^+$ and fixed 84 h later. Actin (red), streptavidin-FITC (green) and merge. Control, electroporated control and electroporated + NAD$^+$. Bar: 10 μm.

Electroporated cells have delayed substrate adherence. Nevertheless, 84 h post-electroporation cells were attached to the substrate and were fixed at that time point. As it is shown in Figure V, there are belt regions which are positive for PAR and biotin, demonstrating NAD$^+$ incorporation to PAR localized in the belt. We understand the results corresponding to the biotin molecule are less striking than the ones for PAR. This might be due to different factors such as a suboptimal cell permeabilization process and the competition of the uptaken biotinylated NAD$^+$ with the more abundant endogenous NAD$^+$ pools, as well the dilution of the labeled NAD+ during the normal cell division process. Taking into account all these considerations, the probability of finding biotinylated NAD$^+$ in the PAR belt is very low. Therefore, appearance of biotin in the PAR belt, though low, should be valued.

We have detected an error in Figure 2. Panel M was a copy of panel N instead of the correspondent merge of panels C and H. We have corrected it.

We thank you for the possibility of improving our work and hope that the manuscript will now be suitable for publication.

Yours faithfully,

Laura Lafon-Hughes and co-authors